# Poplar Biochar as an Alternative Substrate for Curly Endive Cultivated in a Soilless System

**Leo Sabatino** [1] , **Giovanni Iapichino** [1,*], **Rosario Paolo Mauro** [2] ,
**Beppe Benedetto Consentino** [1] **and Claudio De Pasquale** [1]

1   Dipartimento Scienze Agrarie, Alimentari e Forestali (SAAF), University of Palermo, viale delle Scienze, ed. 5-90128 Palermo, Italy; leo.sabatino@unipa.it (L.S.); beppebenedetto.consentino@unipa.it (B.B.C.); claudio.depasquale@unipa.it (C.D.P.)

2   Dipartimento di Agricoltura, Alimentazione e Ambiente (Di3A), University of Catania, via Valdisavoia, 5-95123 Catania, Italy; rosario.mauro@unict.it

*   Correspondence: giovanni.iapichino@unipa.it; Tel.: +39-091-23862215

**Abstract:** Imminent necessity for eco-friendly and low-cost substitutes to peat is a defiance in the soilless plant cultivation systems. Wood biochar could entirely or partly substitute peat as a plant growing constituent to produce vegetables. Nevertheless, knowledge concerning potential plant performance of leafy green vegetables grown on wood biochar is restricted. The present study assessed the main physicochemical traits of various growing media constituted by decreasing the content of peat and by increasing the percentages of poplar wood biochar. Yield, nutritional and functional properties of curly endive plants cultivated in a protected environment were also tested. Biochar was pyrolyzed from poplar (*Populus nigra* L.) at 450 or 700 °C for 48 h. Increasing biochar concentration and pyrolysis temperature resulted in higher pH, EC and K content of the growing mediums. Biochar was also effective in increasing particle density and bulk density. Biochar at 70% and pyrolysis temperature of 450 °C significantly increased head fresh weight by 47.4%, head height by 24.9%, stem diameter by 21.5% and number of leaves by 80.8%, respectively compared with the control (100% peat). Head dry matter content, root dry matter content, SSC, ascorbic acid and total phenolic were also significantly affected by this treatment. Furthermore, the addition of biochar and the use of higher pyrolysis temperature decreased N leaves concentration. This represents a particularly important target for leafy green vegetables healthiness.

**Keywords:** poplar wood biochar; pyrolysis temperature; *Cichorium endivia* L. var. *crispum*; soilless cultivation; substrate mixtures; quality traits

## 1. Introduction

One of the most used substrate constituents for soilless vegetable cultivation is peat [1]. Nevertheless, peat is a restricted resource with a huge demand, and its extraction determines deleterious environmental impacts [2]. Diverse organic resources might occupy an imperative function in lessening the C footprint of the horticulture production by completely or partially replacing peat-based substrates [3]. Biochar is a C-rich organic material, which is produced by thermal decomposition (pyrolysis) of plant-derived biomass in partial or total absence of oxygen [4]. As reported by Verheijen et al. [5], the biomass-based materials are heated to temperature usually between 300 and 1000 °C. Zhang et al. [6] showed that heating time did not modify the properties of biochar, conversely, the pyrolysis temperature has a considerable impact on its physicochemical traits. Thus, compared to other straight biomass-based materials, biochar should go through a series of high temperature cycles to avoid any potential shrinkage of the material. Biochar has also been used as soil amendment in

order to enhance plant growth and yield via increasing water holding capacity, reducing bulk density, increasing cation exchange capacity and soil microbial biomass [7]. Besides, Spokas et al. [8] found that biochar suppress $N_2O$ emissions and ambient $CH_4$ oxidation in laboratory incubations. Chan et al. [9], found that addition of biochar improves N fertilizer use efficiency in radish plants (*Raphanus sativus* var. Long Scarlet) grown in hard-setting Alfisol (Chromosol). Nevertheless, Rogovska et al. [10] report that the use of biochar as a substrate or substrate component is unfavorable due to its high alkalinity and content of leachable polyaromatic hydrocarbons, which are potentially toxic to plants. Biochar can be produced in huge quantities from materials originated from decomposed organic matter, including but not limited to wood and wood materials, bark, grasses and agricultural and industrial by-products [11].

Curly endive (*Cichorium endivia* L. var. *crispum*) is an extensively cultivated leafy green vegetable belonging to the *Asteraceae* family. Curly endive achieved consumers' interest due to its notable quantity of nutritional and functional compounds such as ascorbic acid, phenolics, glucosinolates and sesquiterpene lactones and minerals which are fundamental for humans health and development [12–14].

To our acquaintance, there are no studies regarding the relationship between the pyrolysis temperatures used for biochar production and its use as peat substitute for curly endive production in soilless cultivation. Thus, the aim of our work was to test the main physicochemical traits of various growing media constituted with decreasing content of peat and increasing percentages of poplar wood biochar. The biological results of these substrates on yield, nutritional and functional properties of curly endive plants cultivated in a protected environment were also tested.

## 2. Materials and Methods

### 2.1. Biochar Treatments and Related Analysis

The tested biochar was derived from automatically chipped trunks and large branches of poplar (*Populus nigra* L.) pyrolyzed at 450 or 700 °C for 48 h. The feedstock was free of pollutants and foreign materials. Main physicochemical traits of the used substrates were analyzed. The physicochemical parameters were determined as described by Dispenza et al. [15] and Baiamonte et al. [4]. Thus, the pH was determined with a pH-meter (GLP 21, Crison, Carpi (Modena), Italy, ISO 10390) on a 60 g biochar sample mixed with 300 mL of deionized water after shaking the suspension for 60 min at room temperature (22 °C). The electrical conductivity (EC) was measured in a 1:5 *v/v* substrate–water extract with a conductivity meter (HI 4321, Hanna Instruments, Ronchi di Campanile, Padova, Italy, ISO 11265).

Substrate total nitrogen content (N) was assessed by dry combustion using an elemental analyzer (Carlo Erba Instruments, Milano, Italy, ISO 13878); ash content corrections was used by loss on ignition at 600 °C in an electric muffle furnace.

Substrate phosphorus (P), potassium (K), calcium (Ca), magnesium (Mg) and sodium (Na) were investigated by using 0.2 g of dry sample (105 °C for 24 h) after acid digestion in a microwave oven (CEM Mars Xpress, Matthews, NC, USA); substrate digests were filtered, diluted and analyzed by atomic absorption spectrometry (AAnalyst 200, Perkin-Elmer, Shelton, Connecticut, USA).

Particle density was determined by considering the ratio of organic matter and ash content, taking into account an organic matter density of 1550 kg m$^{-3}$ and an ash density of 2650 kg m$^{-3}$. Dry bulk density was determined considering the ratio of sample mass and the ring volume. Consequently, the total porosity was determined following the physical relationship: total porosity = 1.1—(bulk density/particle density).

Nine growing substrate mixtures (Table 1) were arranged by combining volumes of pelletized biochar (sieved at 5 mm-mesh) and sphagnum peat (0–3 mm sized, H3 decomposition degree, pH 4.3). Biochar and peat were accurately amalgamated in order to minimize breakage of biochar particles, then mixed with 2 L of water and air dried.

**Table 1.** Growing substrates tested for a soilless curly endive cultivation in a protected environment.

| Treatments | Pyrolysis Temperature (°C) | Substrate Mixtures (v:v) |
|---|---|---|
| Peat | _ | 100% peat |
| Biochar 20%_450 | 450 | 80% peat-20% biochar |
| Biochar 20%_700 | 700 | 80% peat-20% biochar |
| Biochar 40%_450 | 450 | 60% peat-40% biochar |
| Biochar 40%_700 | 700 | 60% peat-40% biochar |
| Biochar 70%_450 | 450 | 30% peat-70% biochar |
| Biochar 70%_700 | 700 | 30% peat-70% biochar |
| Biochar 100%_450 | 450 | 100% biochar |
| Biochar 100%_700 | 700 | 100% biochar |

### 2.2. Plant Material and Growing Condition

The trial was carried in an unheated greenhouse covered with polyethylene (PE) at the experimental farm of the Department of Agricultural, Alimentary and Forest Sciences of Palermo (SAAF), at Marsala, Trapani Province (longitude 12°26′ E, latitude 37°47′ N, altitude 37 m) in the north-western coast of Sicily (Italy). Greenhouse temperatures during the growing period were 16–22/12–14 °C (day/night). On 1 February 2018, plantlets at the stage of four to five true leaves of curly endive (*Cichorium endivia* L., var. *crispum* Hegi; var. Trusty, HM Clause, Portes-lès-Valence, France) were transplanted (one plant per pot) into round plastic pots (15 cm in diameter and 13 cm in height) containing 1.0 L of different growing substrate mixtures of peat and poplar wood. The pots were placed in rows 0.5 m apart. In row spacing was 0.25 m (8 plants m$^{-2}$).

The concentrations of all nutrients in the solution introduced into the system were identical for all experimental plots and the composition was as follows: 4.50 mmol L$^{-1}$ of Ca$^{2+}$, 2.00 mmol L$^{-1}$ of H$_2$PO$_4$$^-$, 1.25 mmol L$^{-1}$ NH$_4$$^+$, 1.00 mmol L$^{-1}$ of Mg$^{2+}$, 19.00 mmol L$^{-1}$ of NO$_3$$^-$, 11.00 mmol L$^{-1}$ of K$^+$, 1.10 mmol L$^{-1}$ of SO$_4$$^{2-}$, 40.00 μmol L$^{-1}$ of Fe, 5.00 μmol L$^{-1}$ of Mn, 4.00 μmol L$^{-1}$ of Zn 30.00 μmol L$^{-1}$ of B and 0.75 μmol L$^{-1}$ of Cu [16]. In addition, the irrigation water contained 9.5 mM of Na and 9.0 mM of Cl with an electrical conductivity (EC) of 1.8 mS cm$^{-1}$. The EC and pH in the aforesaid nutrient solution were 2.60 mS cm$^{-1}$ and 5.8, respectively. The pH in the nutrient solution was adjusted to 5.8–5.9 daily by adding appropriate amount of HNO$_3$ water solution. Curly endive plants were managed supplying a complete nutrient solution furnished daily via drip irrigation system. In-line emitters with release rate of 2.0 L h$^{-1}$ were placed (one per pot). As reported by Boztok et al. [17] and Güland Sevigican [18], the quantity of water was predicted according to the solar radiation of the previous day. A leaching fraction of 30% was maintained. The leached nutrient solution was stored in a reservoir tank and not reutilized (open cycle management).

### 2.3. Yield, Nutritional and Functional Properties

Seventy days after transplanting curly endive plants were harvested. Subsequently, head fresh weight, head height, stem diameter and number of leaves were observed on all plants. Head and root dry matter determination, as well as nutritional and functional trait assessment were performed using 5 plants indiscriminately chosen in each replicate. Head dry matter was obtained by drying out of the sample in a heater at 80 °C for the first two days and then dried at 105 °C until constant weight using a thermo-ventilated oven (Memmert, Serie standard, Venice, Italy).

Sampling for the determination of nutritional and functional traits was carried out as reported by Sabatino et al. [14]. Accordingly, all samples were collected subsequent to harvest. A sample of 200 g for each replicate was juiced with a commercial home juicer. The extract was filtered and soluble solids content (SSC) was determined via a digital refractometer (MTD-045nD, Three-In-161 One Enterprises Co. Ltd. Taiwan). Titratable acidity (TA) was assessed by using a potentiometric titration with 0.1 M NaOH up to pH 8.1, 15 mL of plant extract and expressed as percent malic acid equivalents. TA was expressed as a percentage of malic acid [19].

Ascorbic acid was determined by a reflectometer Merck RQflex* 10 m using Reflectoquant Ascorbic Acid Test Strips. Hence, 1 g of leaf juice was dissolved in distilled water, to a total of 10 mL, and mixed. Afterward, a proper test strip was dipped into the sample and inserted into the meter. Results were expressed as mg of ascorbic acid per kg fresh weight.

Total phenolic content was measured by using 5 g of leaf sample, which was weighed out and extracted with methanol and was assayed quantitatively by A765. Total phenolics content was determined as reported by Slinkard and Singleton [20], according to the Folin–Ciocalteu method with slight modifications [21]. The results were expressed as mg of caffeic acid $g^{-1}$ fresh weight.

Leaf total nitrogen (N) was determined by the Kjeldal method. Phosphorus leaves content (P) was determined using colorimetry [22]. Potassium (K), calcium (Ca) and magnesium (Mg) were determined using atomic absorption spectroscopy (SavantAA, 200 ERRECI, Milan, Italy) following wet mineralization [23].

*2.4. Experimental Design and Statistical Analysis*

The experiment was arranged in a randomized complete block design with 3 blocks per treatment and replicated with 10 single plants (9 treatments × 10 plants per treatment = 90 plants per block). Accordingly, the total number of curly endive plants was 270. Data were subjected by one-way analysis of variance using the SPSS software package version 20.0 (StatSoft, Inc., Chicago, IL, USA). For data expressed in percentage, the arcsin transformation before ANOVA analysis ($\varnothing$ = arcsin $(p/100)^{1/2}$) was applied. Tukey HSD test ($p < 0.05$) was applied for means multiple comparisons.

The analysis (principal component analysis—PCA) of the principal components was performed to identify any essential relationship among the different growing substrates, based on the physicochemical substrate traits, morphological, yield, nutritional and functional parameters of curly endive at harvest. Factors with eigenvalues higher than 1.0 were taken in account for the choice of the principal components numbers (PCs). Therefore, the PCs allow the study of correlations between the variables of the input data set. In this regard, the initial variables were projected into the subspace defined by the reduced number of PCs (first and second components) and correlated variables were recognized. The PCA was accomplished using SPSS version 20.0 (StatSoft, Inc., Chicago, IL, USA).

## 3. Results

*3.1. Physicochemical Traits of Growing Substrates*

Data on physicochemical characteristics of the nine growing substrates are presented in Table 2.

**Table 2.** ANOVA analysis and means comparisons for pH, EC, N, P, K, Ca, Mg, Na, total porosity, particle density and bulk density of different growing substrates.

| Treatments | pH | | EC (mS m$^{-1}$) | | N (mg L$^{-1}$) | | P (mg L$^{-1}$) | | K (mg L$^{-1}$) | | Ca (mg L$^{-1}$) | | Mg (mg L$^{-1}$) | | Na (mg L$^{-1}$) | | Total Porosity (% v:v) | | Particle Density (g/L) | | Bulk Density (g/L) | |
|---|---|---|---|---|---|---|---|---|---|---|---|---|---|---|---|---|---|---|---|---|---|---|
| Peat | 5.8 ± 0.1 | h | 12.5 ± 0.3 | h | 100.4 ± 0.8 | a | 30.7 ± 0.6 | a | 100.1 ± 1.3 | g | 104.4 ± 1.1 | a | 38.3 ± 1.0 | a | 18.2 ± 0.2 | a | 89.8 ± 0.3 | bc | 1604.3 ± 1.1 | g | 318.6 ± 1.4 | e |
| Biochar 20%_450 | 6.1 ± 0.1 | g | 15.9 ± 0.3 | g | 68.2 ± 0.9 | b | 18.8 ± 0.4 | b | 117.7 ± 2.4 | f | 72.1 ± 2.1 | b | 18.8 ± 0.3 | b | 15.8 ± 0.4 | b | 90.1 ± 1.4 | bc | 1637.3 ± 2.6 | f | 355.6 ± 6.1 | d |
| Biochar 20% _700 | 6.5 ± 0.1 | f | 16.9 ± 0.4 | f | 64.6 ± 0.5 | c | 18.8 ± 0.2 | b | 124.7 ± 2.1 | e | 71.2 ± 1.0 | b | 18.7 ± 0.6 | b | 15.7 ± 0.3 | b | 94.8 ± 2.5 | a | 1663.6 ± 12.2 | e | 355.3 ± 4.3 | d |
| Biochar 40%_450 | 6.7 ± 0.1 | ef | 16.3 ± 0.2 | fg | 50.7 ± 1.4 | d | 16.7 ± 0.5 | c | 133.4 ± 0.9 | d | 55.4 ± 1.0 | c | 18.1 ± 0.6 | b | 14.4 ± 0.3 | c | 89.4 ± 0.9 | c | 1675.6 ± 3.1 | de | 432.7 ± 2.5 | c |
| Biochar 40%_700 | 7.0 ± 0.1 | e | 18.9 ± 0.1 | e | 47.4 ± 1.0 | e | 16.4 ± 0.2 | c | 137.5 ± 2.0 | cd | 55.3 ± 0.8 | c | 18.1 ± 0.6 | b | 14.4 ± 0.3 | c | 93.7 ± 1.8 | abc | 1682.2 ± 2.6 | d | 432.7 ± 4.4 | c |
| Biochar 70%_450 | 7.9 ± 0.1 | d | 25.8 ± 0.4 | d | 42.2 ± 1.1 | f | 5.5 ± 0.3 | d | 141.9 ± 1.1 | c | 21.2 ± 1.0 | d | 8.1 ± 0.1 | c | 10.0 ± 0.6 | d | 89.4 ± 1.8 | c | 1852.0 ± 1.5 | c | 534.7 ± 2.4 | b |
| Biochar 70%_700 | 8.2 ± 0.1 | c | 28.8 ± 0.3 | c | 39.4 ± 0.4 | g | 5.7 ± 0.2 | d | 153.9 ± 1.6 | b | 21.2 ± 1.0 | d | 8.1 ± 0.2 | c | 10.0 ± 0.7 | d | 94.0 ± 1.7 | ab | 1866.3 ± 2.7 | b | 354.5 ± 1.6 | b |
| Biochar 100%_450 | 8.6 ± 0.1 | b | 35.8 ± 0.1 | b | 37.4 ± 0.4 | gh | 3.8 ± 0.3 | e | 153.0 ± 2.2 | b | 16.5 ± 0.3 | e | 6.2 ± 0.3 | d | 7.5 ± 0.3 | e | 89.9 ± 1.0 | bc | 1872.3 ± 2.0 | b | 649.6 ± 3.8 | a |
| Biochar 100%_700 | 9.1 ± 0.1 | a | 38.3 ± 0.4 | a | 35.2 ± 0.6 | h | 3.7 ± 0.4 | e | 161.2 ± 1.1 | a | 16.4 ± 0.2 | e | 6.4 ± 0.3 | d | 7.3 ± 0.2 | e | 94.0 ± 1.8 | ab | 1887.0 ± 2.4 | a | 650.6 ± 7.6 | a |
| Significance | *** | | *** | | *** | | *** | | *** | | *** | | *** | | *** | | *** | | *** | | *** | |

The same letter within a column identifies data (mean ± standard deviation) not significantly different ($p \leq 0.05$ Tukey HSD Test). Asterisks (***) are designated for the significance ($p$-value $\leq 0.001$).

The highest pH was identified in the growing substrate with 100% biochar produced via a pyrolysis temperature of 700 °C, followed by the growing substrate containing 100% biochar produced via a pyrolysis temperature of 450 °C. The lowest pH value was observed in the substrate with 100% peat. Overall, pH increased (from 5.8 to 9.1) with the increase of biochar content and pyrolysis temperature. A similar behavior was observed for EC, K content and particle density.

ANOVA for N content showed a significant effect of the treatments. The growing substrate with 100% peat had the highest N content, followed by that with 20% biochar produced via a pyrolysis temperature of 450 °C, which, in turn, showed values higher than the growing substrate with the same peat/biochar ratio but produced via a biochar pyrolysis temperature of 700 °C. The combination of biochar at 100% and pyrolysis temperature of 700 °C had the lowest N content.

The growing substrate with 100% peat had the highest P content. Increasing biochar concentration resulted in a P decrement, whereas pyrolysis temperature did not affect P content in the growing substrates tested. Data collected on Ca, Mg and Na supported the trend established for the P content.

The combination of biochar at 20% and pyrolysis temperature of 700 °C had the highest total porosity, whereas, the combinations of biochar at 40% and pyrolysis temperature of 450 °C and that of biochar at 70% and pyrolysis temperature of 450 °C showed the lowest values.

Growing substrate with 100% biochar associated with pyrolysis temperature of 450 or 700 °C had the highest values in terms of bulk density. Decreasing biochar content resulted in a bulk density increase. On the other hand, pyrolysis temperature did not significantly affect the bulk density of the growing substrates tested.

### 3.2. Yield Traits

Data on yield are reported in Table 3.

ANOVA for yield traits revealed a significant effect of the treatments. The combination of biochar at 70% and pyrolysis temperature of 700 °C had the highest values in terms of head fresh weight, followed by the combinations having the same biochar percentage and a pyrolysis temperature of either 450 or 700 °C, which, in turn, showed higher head fresh weight values than biochar 40%_450 growing substrate. Data acquired on head height, stem diameter, number of leaves and root dry matter content supported the tendency established for head fresh weight.

The growing substrate combinations with biochar 40%_700 and biochar 70%_700 °C had the highest head dry matter content, followed by the combination biochar 70%_450. Whereas, the lowest values were obtained in the growing substrate containing 100% peat.

**Table 3.** ANOVA analysis and means comparisons for head fresh weight, head height, stem diameter, number of leaves, head dry matter content and root dry matter content of curly endive cultivated on different growing substrates in a protected environment.

| Treatments | Head Fresh Weight (kg m$^{-2}$) | | Head Height (cm) | | Stem Diameter (mm) | | Number of Leaves (no.) | | Head Dry Matter Content (%) | | Root Dry Matter Content (%) | |
|---|---|---|---|---|---|---|---|---|---|---|---|---|
| Peat | 2.3 ± 0.1 | e | 23.7 ± 0.8 | f | 24.7 ± 0.6 | e | 45.3 ± 2.5 | f | 8.6 ± 0.1 | g | 2.0 ± 0.2 | f |
| Biochar 20%_450 | 2.4 ± 0.1 | de | 25.7 ± 0.4 | de | 25.7 ± 0.6 | de | 55.2 ± 2.3 | e | 9.2 ± 0.3 | f | 2.2 ± 0.1 | de |
| Biochar 20%_700 | 2.6 ± 0.1 | cd | 26.6 ± 0.2 | cde | 27.3 ± 0.6 | cd | 57.8 ± 2.3 | de | 9.8 ± 0.1 | e | 2.4 ± 0.1 | cd |
| Biochar 40%_450 | 2.6 ± 0.1 | c | 27.2 ± 0.3 | c | 27.7 ± 0.6 | bc | 61.3 ± 1.1 | d | 10.3 ± 0.1 | cd | 2.4 ± 0.1 | cde |
| Biochar 40%_700 | 2.9 ± 0.1 | b | 27.7 ± 0.3 | bc | 28.3 ± 0.6 | abc | 68.1 ± 1.0 | bc | 10.9 ± 0.1 | a | 2.5 ± 0.1 | bc |
| Biochar 70%_450 | 3.0 ± 0.1 | b | 28.5 ± 0.4 | ab | 29.3 ± 0.6 | ab | 71.4 ± 1.2 | b | 10.5 ± 0.1 | bc | 2.7 ± 0.1 | ab |
| Biochar 70%_700 | 3.3 ± 0.1 | a | 29.6 ± 0.6 | a | 30.0 ± 0.6 | a | 81.9 ± 2.7 | a | 10.9 ± 0.2 | a | 2.8 ± 0.1 | a |
| Biochar 100%_450 | 2.4 ± 0.1 | e | 25.6 ± 0.6 | e | 25.3 ± 0.6 | e | 58.9 ± 1.2 | de | 10.0 ± 0.2 | de | 2.2 ± 0.1 | e |
| Biochar 100%_700 | 2.6 ± 0.1 | cd | 26.9 ± 0.1 | cd | 27.7 ± 1.2 | bc | 63.0 ± 2.0 | cd | 10.4 ± 0.1 | c | 2.4 ± 0.1 | cde |
| Significance | *** | | *** | | *** | | *** | | *** | | *** | |

The same letter within a column identifies data (mean ± standard deviation) not significantly different ($p \leq 0.05$ Tukey HSD Test). Asterisks (***) are designated for the significance ($p$-value $\leq 0.001$).

### 3.3. Nutritional and Functional Properties

Combining biochar at 70% and pyrolysis temperature at 450 °C resulted in the highest SSC. The lowest SSC were recorded from plants cultivated in the growing substrate with biochar at 100% and a pyrolysis temperature of 700 °C (Table 4). Overall, SSC decreased with the increase of the pyrolysis temperature.

**Table 4.** Analysis of variance and means comparisons for SSC, TA, ascorbic acid and total phenolic of curly endive cultivated on different growing substrates in a protected environment.

| Treatments | SSC (°Brix) | | TA (%) | | Ascorbic Acid (mg kg$^{-1}$ f.w.) | | Total Phenolic (mg of Caffeic Acid g$^{-1}$ f.w.) | |
|---|---|---|---|---|---|---|---|---|
| Peat | 4.3 ± 0.1 | cd | 0.7 ± 0.1 | a | 69.5 ± 3.3 | e | 0.57 ± 0.04 | ef |
| Biochar 20%_450 | 4.4 ± 0.1 | bc | 0.6 ± 0.2 | a | 83.3 ± 2.7 | d | 0.64 ± 0.02 | de |
| Biochar 20%_700 | 4.2 ± 0.1 | d | 0.6 ± 0.1 | a | 81.7 ± 2.9 | d | 0.53 ± 0.01 | f |
| Biochar 40%_450 | 4.6 ± 0.1 | b | 0.6 ± 0.1 | a | 90.6 ± 1.5 | bc | 0.76 ± 0.02 | bc |
| Biochar 40%_700 | 4.3 ± 0.1 | cd | 0.6 ± 0.1 | a | 85.9 ± 1.5 | cd | 0.67 ± 0.03 | cd |
| Biochar 70%_450 | 4.8 ± 0.1 | a | 0.7 ± 0.1 | a | 103.2 ± 2.1 | a | 0.92 ± 0.03 | a |
| Biochar 70%_700 | 4.6 ± 0.1 | b | 0.7 ± 0.2 | a | 96.1 ± 0.9 | b | 0.88 ± 0.04 | b |
| Biochar 100%_450 | 4.3 ± 0.1 | cd | 0.7 ± 0.1 | a | 87.1 ± 2.7 | cd | 0.95 ± 0.04 | a |
| Biochar 100%_700 | 4.2 ± 0.1 | d | 0.7 ± 0.1 | a | 82.0 ± 2.3 | d | 0.81 ± 0.02 | b |
| Significance | *** | | NS | | *** | | *** | |

The same letter within a column identifies data (mean ± standard deviation) not significantly different ($p \leq 0.05$ Tukey HSD Test). Asterisks (***) are designated for the significance ($p$-value $\leq 0.001$).

ANOVA for TA did not show a significant effect of the treatments (Table 4). Data collected on ascorbic acid supported the trend established for the SSC (Table 4). Curly endive plants cultivated on biochar at 70% with a pyrolysis temperature of 450 °C and on biochar at 100% with a pyrolysis temperature of 450 °C had the highest total phenolic content (Table 4), followed by those grown on biochar at 70% with a pyrolysis temperature of 700 °C and biochar at 100% with a pyrolysis temperature of 700 °C. The lowest total phenolic content was recorded from plants cultivated on biochar at 20% with a pyrolysis temperature of 700 °C. However, plants grown on peat at 100% did not significantly differ from plants grown on biochar at 20% with a pyrolysis temperature of 450 °C and from those grown on biochar 20% with a pyrolysis temperature of 700 °C.

Data on plant minerals content are presented in Table 5. ANOVA for P and Mg content did not reveal a significant effect of the treatments.

**Table 5.** Analysis of variance and means comparisons for leaves N, P, K, Ca and Mg content of curly endive cultivated on different growing substrates in a protected environment.

| Treatments | N (mg g$^{-1}$ DW) | | P (mg g$^{-1}$ DW) | | K (mg g$^{-1}$ DW) | | Ca (mg g$^{-1}$ DW) | | Mg (mg g$^{-1}$ DW) | |
|---|---|---|---|---|---|---|---|---|---|---|
| Peat (PE) | 6.42 ± 0.11 | a | 0.63 ± 0.03 | a | 2.92 ± 0.03 | f | 0.75 ± 0.03 | b | 0.37 ± 0.01 | a |
| Biochar 20%_450 (BC24) | 4.85 ± 0.10 | b | 0.63 ± 0.01 | a | 3.07 ± 0.06 | e | 0.68 ± 0.02 | cd | 0.35 ± 0.02 | a |
| Biochar 20%_700 (BC27) | 6.66 ± 0.08 | bc | 0.62 ± 0.01 | a | 3.21 ± 0.02 | d | 0.82 ± 0.02 | a | 0.35 ± 0.01 | a |
| Biochar 40%_450 (BC44) | 4.44 ± 0.13 | cd | 0.63 ± 0.03 | a | 3.30 ± 0.06 | d | 0.62 ± 0.02 | ef | 0.35 ± 0.01 | a |
| Biochar 40%_700 (BC47) | 4.27 ± 0.10 | de | 0.62 ± 0.03 | a | 3.44 ± 0.02 | c | 0.72 ± 0.02 | bc | 0.35 ± 0.01 | a |
| Biochar 70%_450 (BC74) | 4.13 ± 0.08 | e | 0.62 ± 0.03 | a | 3.65 ± 0.03 | b | 0.55 ± 0.01 | g | 0.33 ± 0.01 | a |
| Biochar 70%_700 (BC77) | 3.81 ± 0.10 | f | 0.62 ± 0.01 | a | 3.71 ± 0.03 | b | 0.62 ± 0.01 | de | 0.36 ± 0.01 | a |
| Biochar 100%_450 (BC14) | 3.52 ± 0.08 | g | 0.64 ± 0.02 | a | 3.87 ± 0.02 | a | 0.42 ± 0.03 | h | 0.35 ± 0.01 | a |
| Biochar 100%_700 (BC17) | 3.23 ± 0.12 | h | 0.64 ± 0.03 | a | 3.96 ± 0.02 | a | 0.56 ± 0.02 | fg | 0.35 ± 0.02 | a |
| Significance | *** | | NS | | *** | | *** | | NS | |

The same letter within a column identifies data (mean ± standard deviation) not significantly different ($p \leq 0.05$ Tukey HSD Test). Asterisks (***) are designated for the significance ($p$-value $\leq 0.001$).

Curly endive plants grown on 100% peat had the highest leaves N content, followed by those cultivated on a substrate mixture with 20% biochar produced via a pyrolysis temperature of 450 °C,

which in turn showed values higher than plants grown on biochar at 20% with a pyrolysis temperature of 700 °C. However, ANOVA, did not reveal statistical significant differences between biochar at 20% with a pyrolysis temperature of 450 °C and biochar at 20% with a pyrolysis temperature of 700 °C in terms of N leaf content. The lowest leaf N content was detected in plants from biochar 100% with a pyrolysis temperature of 700 °C.

Plants cultivated on 100% biochar growing substrates produced both with a pyrolysis temperature of 450 and 700 °C had the highest values in terms of leaves K content, followed by those grown on substrate mixtures with 70% of biochar (both produced with a pyrolysis temperature of 450 or 700 °C). Plants from plots with 100% peat revealed the lowest leaves K values. However, in general, K concentration increased with the increase of biochar content.

The highest leaves Ca contents were observed in plants grown on biochar at 20% with a pyrolysis temperature of 700 °C, while the lowest values were detected in plants grown on biochar at 100% with a pyrolysis temperature of 450 °C.

### 3.4. PCA Analysis

The findings of the principal component analysis (PCA) displayed four main components (PCs) with eigenvalues higher than 1.00 (Table 6), accounting for 62.58%, 21.59%, 9.19% and 4.14% of the total variance, respectively.

**Table 6.** Loadings scores for principal components PC1, PC2, PC3 and PC4.

| Variable | PC1 | PC2 | PC3 | PC4 |
|---|---|---|---|---|
| pH | **0.922** | −0.307 | −0.201 | 0.077 |
| EC | **0.851** | −0.444 | −0.247 | 0.053 |
| $N_{substrate}$ * (N-$s$) | **−0.957** | −0.049 | 0.087 | 0.168 |
| $P_{substrate}$ * (P-$s$) | **−0.981** | 0.130 | 0.026 | 0.087 |
| $K_{substrate}$ * (K-$s$) | **0.963** | −0.086 | −0.225 | −0.002 |
| $Ca_{substrate}$ * (Ca-$s$) | **−0.991** | 0.124 | −0.001 | 0.027 |
| $Mg_{substrate}$ * (Mg-$s$) | **−0.954** | 0.027 | 0.055 | 0.237 |
| $Na_{substrate}$ * (Na-$s$) | **−0.932** | 0.345 | 0.077 | −0.008 |
| Total porosity (TP) | 0.210 | 0.434 | **−0.863** | 0.038 |
| Particle density (PD) | **0.937** | −0.243 | −0.038 | 0.130 |
| Bulk density (BD) | **0.896** | −0.425 | −0.099 | 0.047 |
| Head fresh weight (HFW) | 0.592 | **0.745** | 0.013 | 0.303 |
| Head height (HH) | **0.745** | **0.655** | 0.045 | 0.071 |
| Stem diameter (SD) | **0.659** | **0.711** | 0.025 | 0.122 |
| Number of leaves (NL) | **0.790** | 0.571 | 0.002 | 0.207 |
| Head dry matter content (HDMC) | **0.831** | 0.446 | −0.107 | 0.038 |
| Root dry matter content (RDMC) | **0.692** | **0.703** | 0.075 | 0.080 |
| SSC | 0.352 | 0.366 | **0.846** | 0.115 |
| TA | 0.458 | **−0.678** | 0.245 | 0.490 |
| Ascorbic acid (AA) | **0.763** | 0.404 | 0.488 | −0.090 |
| Total Phenolic (TPC) | **0.837** | −0.377 | 0.377 | 0.030 |
| $N_{leaves}$ ** (N-$l$) | **−0.929** | 0.064 | 0.233 | 0.224 |
| $P_{leaves}$ ** (P-$l$) | 0.034 | **−0.958** | 0.047 | 0.105 |
| $K_{leaves}$ ** (K-$l$) | **0.947** | −0.242 | −0.187 | 0.036 |
| $Ca_{leaves}$ ** (Ca-$l$) | **−0.726** | 0.584 | −0.324 | −0.005 |
| $Mg_{leaves}$ ** (Mg-$l$) | **−0.604** | −0.062 | −0.305 | **0.683** |
| Eigenvalue | 16.270 | 5.613 | 2.389 | 1.077 |
| Variance % | 62.576 | 21.589 | 9.189 | 4.140 |
| Cumulative % | 62.576 | 84.165 | 93.354 | 97.494 |

Values in bold within the same factor indicate the variable with the largest correlation. * $N_{substrate}$, substrate nitrogen content; $P_{substrate}$, substrate phosphorus content; $K_{substrate}$, substrate potassium content; $Ca_{substrate}$, substrate calcium content; $Mg_{substrate}$, substrate magnesium content; $Na_{substrate}$, substrate sodium content. ** $N_{leaves}$, leaves nitrogen content; $P_{leaves}$, leaves phosphorus content; $K_{leaves}$, leaves potassium content; $Ca_{leaves}$, leaves calcium content; $Mg_{leaves}$, leaves magnesium content.

This evidenced that the twenty-six variables could be articulated as a linear arrangement of four PCs, elucidating 97.50% of the total variance. PC1 was mainly positively correlated to pH, EC, $K_{substrate}$, particle density, bulk density, head height, stem diameter, number of leaves, head dry matter content, root dry matter content, ascorbic acid, total phenolic, $K_{leaves}$ and mainly negatively correlated to $N_{substrate}$, $P_{substrate}$, $Ca_{substrate}$, $Mg_{substrate}$, $Na_{substrate}$, $N_{leaves}$, $Ca_{leaves}$ and $Mg_{leaves}$; PC2 was mainly positively related to head fresh weight, head height, stem diameter, root dry matter content, and negatively related to $P_{leaves}$; PC3 was mainly positively correlated to SSC, and negatively related to total porosity; and PC4 was mainly positively related to $Mg_{leaves}$ (Table 6).

The representation of the dependent variables on the plane PC1-PC2- provides such a relationship, as showed in the plot of loadings (Figure 1).

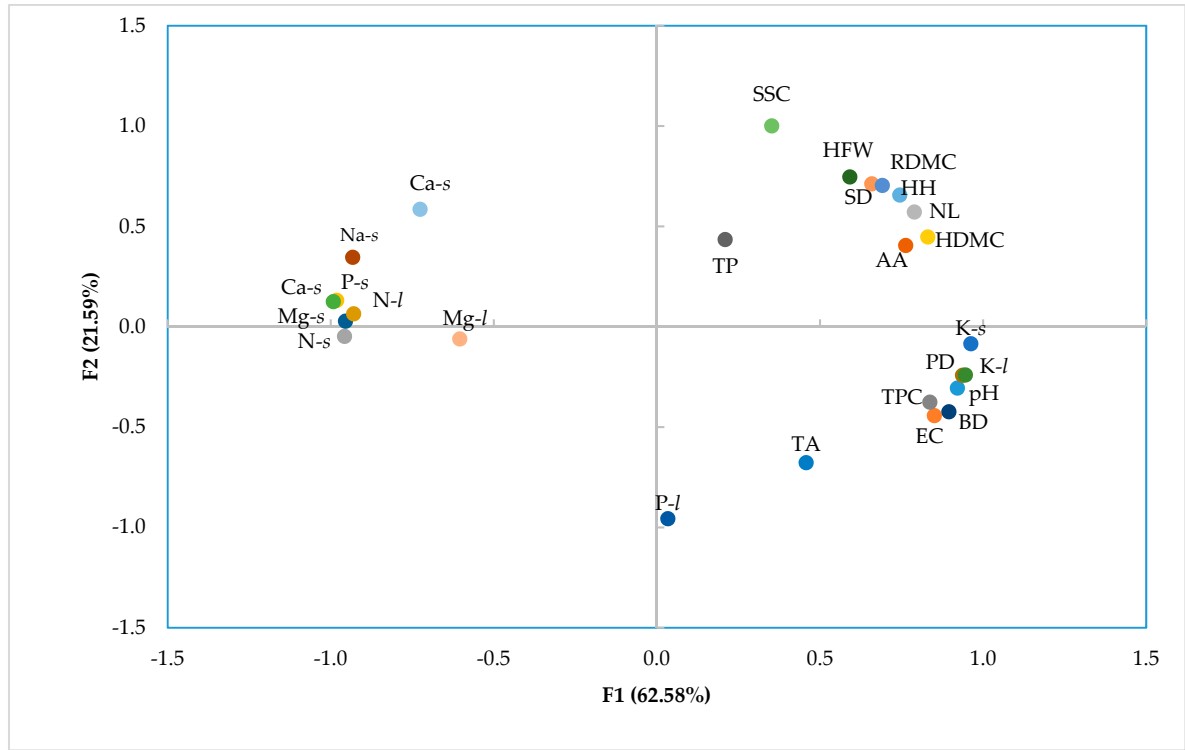

**Figure 1.** Plot of loadings: substrate traits, agronomic, nutritional and functional properties of curly endive at harvest.

The discrimination of the various growing substrates can be deduced in the plot of scores (Figure 2), where four clusters could be defined.

The peat and biochar 20%_450 are placed on the bottom-left side of the plot of loading, the biochar 20%_700, biochar 40%_450 and biochar 40%_700 are allocated on the top-left side of the plot of loading. Whereas, other samples are situated on its right side, with biochar 100%_450 and biochar 100%_700 sited in the bottom-right position (Figure 2).

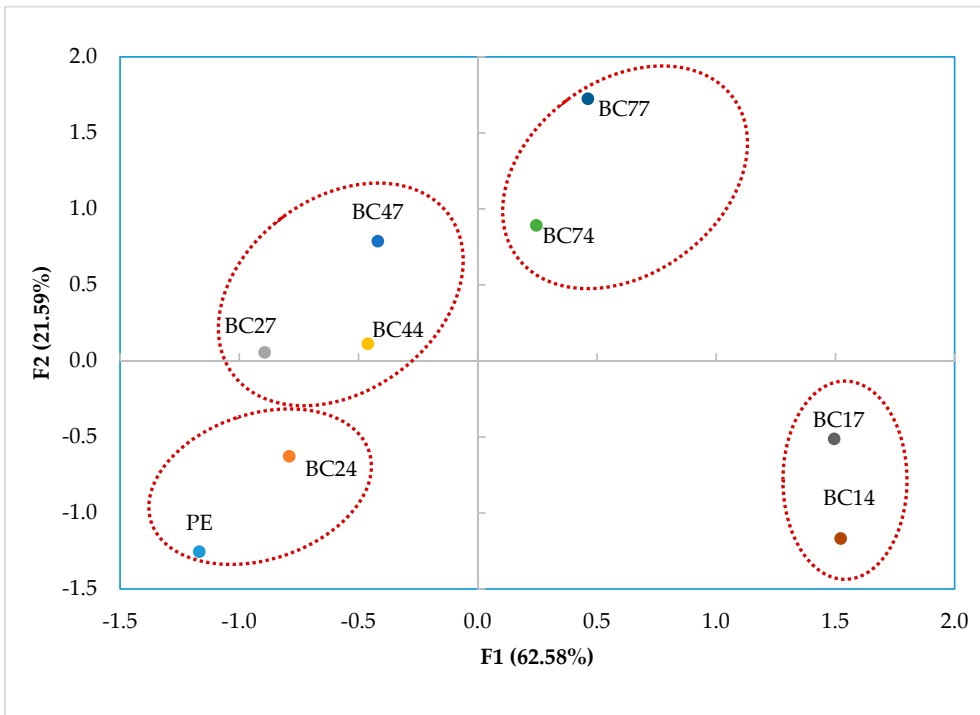

**Figure 2.** Plot of scores (trial) produced by PC1 and PC2 of the principal component analysis (PCA) analysis.

## 4. Discussion

In this study, we assessed the potential employ of diverse peat/biochar-based mixtures as growing substrates for curly endive grown in a protected environment.

Results on pH confirmed those by Dispenza et al. [15] who evidenced lower pH values in a peat-based substrate compared to the peat/biochar mixtures or biochar-based growing substrates. Our findings are also similar to those by Tian et al. [24], who evidenced lower pH levels in a peat based growing medium than in green waste biochar and in their combination. Our outcomes on pH values are, furthermore, in accord with those reported by Choi et al. [11], who by inquiring the influence of biochar combinations with pine-bark based growing media on growth and development of horticultural crops, found that 100% biochar addition increased the substrate pH. In the present study, the pH detected in those growing media, which included up to 70% of biochar were in the 6.1–8.2 range. This range is convenient for curly endive [25]. Results on pH revealed that pH values increased with the increase of the pyrolysis temperature. This is in accord with the results of Conz et al. [26] who, studying the influence of pyrolysis temperature and feedstock type on agricultural properties and stability of biochars, found that, overall, pH increased with the increase of the pyrolysis temperature (ranging from 350 to 650 °C). Results concerning EC demonstrated that both increased the biochar percentage in the growing substrate and pyrolysis temperature increase EC values. Similarly Dispenza et al. [15] found that substrate EC increased with the increase of biochar content (from 12 to 36 mS m$^{-1}$ for 100% peat and 100% biochar, respectively) when biochar substituted peat in the growing medium used for *Euphorbia × lomi*. Moreover our findings on EC are in accord with those obtained by Kloss et al. [27], Oh et al. [28] and Song and Guo [29], who found increased EC with the pyrolysis temperature increase. Higher EC values feature higher soluble salt availability, which in turn, presumably, accounted for the higher plant performance evidenced in this study. Nevertheless, our study is in contrast with the report of Conz et al. [26], who found that increasing pyrolysis temperature did not increase EC. In this regard, we could hypothesize that this different result is due to a different type of raw materials used for biochar production. Our outcomes are partially conform to those by Dispenza et al. [15] who found higher values of N in the medium with 100% peat and affirmed that the N content had no significant variation

among the growing media amended with (from BC20 to BC100). In the present study, P concentrations dropped as biochar percentage increased. However, pyrolysis temperature did not influence substrate P content. Our outcomes are in line with those experienced by Karami et al. [30], who testing green waste compost and biochar soil amendments in order to reduce lead and copper mobility and uptake to ryegrass, report that P availability was lower in a soil amended with biochar than in green waste compost-amended soil. Furthermore, results are in accord with those of Altland and Locke [31] who, inquiring how biochar type might affect macronutrient retention and release from soilless media, evidenced that increasing levels of biochar will result in a substantial increase of K in the substrate and should be considered for fertility programs although being a modest source of P for ornamental plant production. Our outcomes on P content are, also, in accord with those by Conz et al. [26] who evidenced equal P concentration with the increase pyrolysis temperature for sawdust and rice husk biochars. Results on K content confirmed that, both increased biochar percentage in the growing substrate and pyrolysis temperature increase K concentrations. Others have reported that K content increases with higher levels of biochar (Dispenza et al.) [15]. Moreover, our data are in agreement with those by Bedussi et al. [32] who found that the addition of spruce wood biochar to peat allowed the preservation of stable and high levels of K in the pore water, both in the root free substrate and in the rhizosphere in planted and non-planted soilless growing media. Nonetheless, our results on K concentration are in contrast to Conz et al. finding [26] who report no influence of the pyrolysis temperatures on substrate K concentrations for sugarcane straw, rice husk, poultry litter and sawdust biochars. Outcomes displayed that Ca, Mg and Na contents decreased with the biochar percentage increase in the growing media. Furthermore, pyrolysis temperature did not influence Ca, Mg and Na concentrations. This is confirmed by Dispenza et al. [15]. Additionally, our findings agree with Conz et al. [26], who affirm that pyrolysis temperature has no effect on Mg concentration. Our work highlighted that increasing poplar biochar percentage in the growing substrates corresponds with an increase of particle density and bulk density. These results are supported by several authors [11,15,24,33]. Outcomes on yield traits pointed out that an increase of poplar biochar percentage (until 70% of the total volume) and pyrolysis temperature (700 °C) can improve curly endive crop performance. This is probably because biochar can retain high amounts of exchange cations due to its high porosity and surface/volume ratio and, consequently, can enhance plant nutrients uptake and availability [9,34,35]. Furthermore, we found that increasing biochar percentage results in an increase of biomass. Similarly, Tian et al. [24] refer that mixing green waste biochar with peat provides a better physical environment than biochar alone or peat alone in growing media for *Calathea rotundifola* cv. Fasciata. However, our data remarked that the lowest plant yield performance was recorded from plants grown on 100% biochar or 100% peat. This is probably due to the decline of growing medium physicochemical traits such as high pH and EC levels. Our results agree with those of Dispenza et al. [15] and Rondon et al. [36] who, by studying the biological nitrogen fixation in common beans (*Phaseolus vulgaris* L.) with bio-char additions, reported that adding a high rate of Eucalyptus wood biochar to a poor soil in a soilless study crop performance possibly due to a micronutrient shortage caused by an augmentation of soil pH.

Outcomes on total phenolic revealed that the highest values were found in plants from biochar 70%_450 and biochar 100%_450 treatments. These findings are supported by those of Zulfiqar et al. [37] who, by inquiring the effects of the amendment of a biochar (BC) or a biochar-compost mixture (BioComp) to a peat-based substrate at 20% by volume on the growth of *Syngonium podophyllum*, found that total phenolic contents were higher in plants grown in BC- or BioComp-amended media. In our study, the lowest curly endive crop performance was observed in plants cultivated in a growing substrate with 100% biochar. This remarks that the unfavorable plant growing conditions of the media and the stressful conditions induce an accumulation of phenolics [38–41]. Therefore, we could hypothesize that the higher total phenolic content is due to a stress caused by a non-optimal curly endive-cultivation environment (pH and EC stresses). Our study revealed, also, that the higher SSC and ascorbic acid contents were recorded in curly endive plants grown on biochar at 70% and a pyrolysis temperature of 450 °C and that TA was not affected by the growing substrates tested. Our results

highlighted a decrease of N content and an increase of K in leaves as the biochar content in the growing mediums increased. The highest leaf Ca concentrations were observed in plants from biochar at 20% and pyrolysis temperature of 700 °C. Whereas, other minerals such as P and Mg were not influenced by the growing media. The foliar mineral contents were in the range that permits a satisfactory plant growth and development in agreement with the average concentration of mineral nutrients in plant dry matter. As reported by Awad et al. [42], the release of nutrients from biochar-based material may be one of the likely mechanisms for improving K, Mg, Mn and Zn uptake by plant root systems in perlite + rice husk biochar substrate. In particular, Awad et al. [42] found that rice husk biochar induced the growth of beneficial microorganisms on its surfaces. This may enhance the uptake of nutrients by plant. Furthermore, comparable mechanism for improving the uptake of macronutrients by maize plants in soil treated with biochar were described by Kim et al. [43], Lee et al. [44] and Rehman et al. [45]. In our study, even though a higher amount of leaves K content, regularly implicated in the photosynthetic process, is positively correlated to percentage of biochar in the growing media, the highest leaves K levels were collected in plants grown on substrates with biochar at 70%. The explanation for this might be due to the fact that 100% biochar substrates caused a plant stress due to non-optimal pH and EC. Our results on Ca are similar to those observed by Zhang et al. [46] who, investigating the effects of the biochar and humic acid amendments on the quality of composted green waste found that leaves nutrients content significantly increased when *Calathea insignis* plants were grown in media consisted of composted green waste and 20% coir biochar. However, as reported by Conz et al. [26], a large variability on plants nutrient uptake depends on the origin of biochar-based materials used for biochar production [26].

## 5. Conclusions

In the present study, biochar percentage and pyrolysis temperature significantly affected yield, nutritional and functional traits of curly endive plants. Biochar-based substrates amended with peat improved crop performance. Particularly, combining 70% of biochar with a pyrolysis temperature of 450 or 700 °C successfully enhanced yield traits, SSC, ascorbic acid and total phenolic as compared with the control (100% peat) and at the same time reduced leaves N content. This is particularly important for leafy vegetables healthiness. However, our results also showed that the use of poplar biochar alone is not optimal for curly endive plants due to its high pH and EC levels. Finally, as the vegetable production sector is a high consumer of peat, findings from the present work highlight that poplar biochar based-substrates obtained by a pyrolysis temperature of 450 or 700 °C may be precious replacements of peat-based media in soilless leafy vegetables production, consequently contributing to peatlands safeguarding and maintenance.

**Author Contributions:** Conceptualization, L.S. and G.I.; methodology, L.S.; validation, L.S., G.I. and F.D.; formal analysis, L.S.; investigation, B.B.C.; resources, C.D.P.; data curation, L.S. and R.P.M.; writing—original draft preparation, L.S.; writing—review and editing, L.S and G.I.; supervision, L.S. All authors have read and agreed to the published version of the manuscript.

**Funding:** This research received no external funding.

**Conflicts of Interest:** The authors declare no conflict of interest.

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
