# Peer review of "Poplar Biochar as an Alternative Substrate for Curly Endive Cultivated in a Soilless System"

_applsci, doi:10.3390/app10041258_

Round 1

Reviewer 1 Report

Dear author and editor,

in my opinion the manuscript “Poplar biochar as alternative substrate for curly endive cultivated in a soilless system” is acceptable for publication on Applied Sciences Journal after some minor revisions.

Please find below my specific comments divided per section.

Abstract: clear and concise

Introduction: clear and informative.

Line 58: there is something missing: extensively cultivated? Consumed? Please add

Materials and methods:

I suggest to reorganize this chapter with a different division of subchapters. For example, as biochar treatments are investigated as your experiment results, I would divide biochar description and related analyses from the plant material and growing conditions description. A new subchapter list could be:

2.1 biochar treatments; 2.2 plant material and growing conditions; 2.3 experimental design; 2.4 yield, nutritional and functional properties; 2.5 statistical analyses

Line 107: please specify here how many replicas per treatment you did use

Line 114-115: why does your irrigation water contain NaCl? If it is a saline source please provide also its EC value

Line 129: in each treatment instead then in each replicate

Line 154-155:I have not clear why you consider 3 replicas made of 10 pots each: why not considering 30 replicas per treatment?

Results:

Tables 2, 3, 4, 5: please present in all tables together with your results also the standard deviation or the error mean

Discussion:

Clear and well referenced

Author Response

REVIEWER 1

Abstract: clear and concise

Introduction: clear and informative.

Point 1. Line 58: there is something missing: extensively cultivated? Consumed? Please add

Response 1. The phrase was modified as requested.

Materials and methods:

Point 2. I suggest to reorganize this chapter with a different division of subchapters. For example, as biochar treatments are investigated as your experiment results, I would divide biochar description and related analyses from the plant material and growing conditions description. A new subchapter list could be:

2.1 biochar treatments; 2.2 plant material and growing conditions; 2.3 experimental design; 2.4 yield, nutritional and functional properties; 2.5 statistical analyses

Response 2. The Materials and Methods section was reorganized taking into account the reviewer's advises.

Point 3. Line 107: please specify here how many replicas per treatment you did use

Response 3.The number of replicates per treatment used was specified in the dedicated section "2.4 Experimental design and statistical analysis".

Point 4.Line 114-115: why does your irrigation water contain NaCl? If it is a saline source please provide also its EC value

Response 4.The irrigation water contains NaCl, as reported. The irrigation water EC value was added.

Point 5.Line 129: in each treatment instead then in each replicate

Response 5.Actually, we used 5 plants, randomly chosen, per replicate and not per treatment. Consequently, we can't modify the sentence.

Point 6.Line 154-155:I have not clear why you consider 3 replicas made of 10 pots each: why not considering 30 replicas per treatment?

Response 6.The experimental design was better explained.

Results:

Point 7.Tables 2, 3, 4, 5: please present in all tables together with your results also the standard deviation or the error mean

Response 7.The standard deviation was added to our results, as requested.

Discussion:

Clear and well referenced

Reviewer 2 Report

In this manuscript, different biochar percentages and pyrolysis temperatures are set. The physical and chemical properties of each mixture, the growth state of curly endive in different mixtures, the main active ingredient content of curly endive, the element content in curly endive leaves, and principal component analysis are determined. Studies have shown that the percentage of biochar and pyrolysis temperature significantly affect the yield, nutritional and functional traits of curly chicory plants. Biochar-based substrates amended with peat improved crop performance. The manuscript has a certain innovative, but there are still some problems inadequacies should be improved, such as: 1. When evaluating the influence of the cultivation substrate on the growth status of curly endive, why the N, P, K, Ca and Mg ions in the leaves were measured, and whether the above-mentioned ions content have a greater impact on the quality of curly endive, and it should be explained in the introduction. 2. When evaluating the influence of the cultivation substrate on the growth of curly endive, in addition to the growth state and active substance content of curly endive, I think the growth rate and yield per unit area of curly endive are also very important indicators. 3. The use of biochar requires high temperature treatment. Increasing the temperature will consume resources and cause pollution. How can it be proved that the mixed use of biochar and peat is more economical? 4. The discussion should focus on the effects of mixed substrates on the growth of curly endive, the advantages and disadvantages of mixed substrates, the direction of improvement, etc., rather than the consistency of their experimental results with their predecessors. The entire discussion should be carefully reviewed and revised.

Author Response

REVIEWER 2

Point 1. When evaluating the influence of the cultivation substrate on the growth status of curly endive, why the N, P, K, Ca and Mg ions in the leaves were measured, and whether the above-mentioned ions content have a greater impact on the quality of curly endive, and it should be explained in the introduction.

Response 1. Why the minerals in the leaves were measured was specified in the introduction as suggested.

Point 2. When evaluating the influence of the cultivation substrate on the growth of curly endive, in addition to the growth state and active substance content of curly endive, I think the growth rate and yield per unit area of curly endive are also very important indicators.

Response 2. The yield was expressed as kg m-2 as requested.

Point 3. The use of biochar requires high temperature treatment. Increasing the temperature will consume resources and cause pollution. How can it be proved that the mixed use of biochar and peat is more economical?

Response 3. Biochar environmental sustainability is fundamentally related to the conversion of locally available waste biomass materials into biochar for soil or soilless agricultural applications [Alshankiti A. and S. Gill. 2016. Integrated plant nutrient management for sandy soil using chemical fertilizers, compost, biochar and biofertilizers-Case study in UAE. Journal of arid land studies, 26-3,101-106 (2016)]. Moreover, biochar can contribute to a better water storage by modifying the substrate pore size distribution (Downie, A., Crosky A., Munroe P. Physical properties of biochar. In: Lehmann, J., S. Joseph (eds), Biochar for environmental management: Science and Technology, Earthscan, London, 2009). Nevertheless, the organic carbon produced in biochar is very stable. Moreover, biochar is more stable at higher temperature. The use of biochar in the soilless systems has the potential to both improve the vegetable production and sequester carbon, which is important for mitigation of excessive carbon dioxide in the atmosphere (McHenry, M.P. Agricultural bio-char production, renewable energy generation and farm carbon sequestration in Western Australia: Certainty, uncertainty and risk. Agric. Ecosyst. Environ. 2009, 129, 1–7). Therefore, the application of higher temperature for biochar production have been considered for a suitable carbon cycle storage and water agricultural uses.

Point 4. The discussion should focus on the effects of mixed substrates on the growth of curly endive, the advantages and disadvantages of mixed substrates, the direction of improvement, etc., rather than the consistency of their experimental results with their predecessors. The entire discussion should be carefully reviewed and revised. 

Response 4.  In our opinion most of the suggestions concerning Point 4 were already incorporated in the Conclusion section. However, the discussion section was also improved as requested.